# Assessment of Insecticidal Activity of Benzylisoquinoline Alkaloids from Chilean Rhamnaceae Plants against Fruit-Fly *Drosophila melanogaster* and the Lepidopteran Crop Pest *Cydia pomonella*

**DOI:** 10.3390/molecules25215094

**Published:** 2020-11-03

**Authors:** Soledad Quiroz-Carreño, Edgar Pastene-Navarrete, Cesar Espinoza-Pinochet, Evelyn Muñoz-Núñez, Luis Devotto-Moreno, Carlos L. Céspedes-Acuña, Julio Alarcón-Enos

**Affiliations:** 1Laboratorio de Síntesis y Biotransformación de Productos Naturales, Dpto. Ciencias Básicas, Universidad del Bio-Bio, PC3780000 Chillán, Chile; sole.m.quiroz.c@gmail.com (S.Q.-C.); edgar.pastene@gmail.com (E.P.-N.); evdmunoz@gmail.com (E.M.-N.); cespedes.leonardo@gmail.com (C.L.C.-A.); 2Dpto. Agroindustria, Facultad de Ingeniería Agrícola, Universidad de Concepción, 3780000 Chillán, Chile; espinoza.cesar59@gmail.com; 3Instituto de Investigaciones Agropecuarias, INIA Quilamapu, 3780000 Chillán, Chile; ldevotto@inia.cl

**Keywords:** botanical insecticides, plant-insect interaction, aporphines, tetra-hydro-isoquinolines, octopamine-receptor, ecdysone-receptor

## Abstract

The Chilean plants *Discaria chacaye*, *Talguenea quinquenervia* (Rhamnaceae), *Peumus boldus* (Monimiaceae), and *Cryptocarya alba* (Lauraceae) were evaluated against Codling moth: *Cydia pomonella* L. (Lepidoptera: Tortricidae) and fruit fly *Drosophila melanogaster* (Diptera: Drosophilidae), which is one of the most widespread and destructive primary pests of *Prunus* (plums, cherries, peaches, nectarines, apricots, almonds), pear, walnuts, and chestnuts, among other. Four benzylisoquinoline alkaloids (coclaurine, laurolitsine, boldine, and pukateine) were isolated from the above mentioned plant species and evaluated regarding their insecticidal activity against the codling moth and fruit fly. The results showed that these alkaloids possess acute and chronic insecticidal effects. The most relevant effect was observed at 10 µg/mL against *D. melanogaster* and at 50 µg/mL against *C. pomonella*, being the alteration of the feeding, deformations, failure in the displacement of the larvae in the feeding medium of *D. melanogaster*, and mortality visible effects. In addition, the docking results show that these type of alkaloids present a good interaction with octopamine and ecdysone receptor showing a possible action mechanism.

## 1. Introduction

Prunus are one of the main commodities produced in the agricultural business in Chile. The high demand for these products has driven an increase in the cultivated area year by year, which added to the need to expand the diversity of crops and meet the demands of organic and ecological production, and have motivated the need to search for new environmentally-friendly bio-pesticides [1].

These commodities suffer the attack of different insect pests being *C. pomonella* (L.) (Lepidoptera: Tortricidae) as one of the most recurrent during different stages of pruning the fruits, which can cause many quantitative and qualitative losses, and can negatively affect the safety of fresh fruits [2]. *D. melanogaster Meigen*. (Diptera: Drosophilidae) also called vinegar fly or fruit fly. It is a species of brachyderus diptera of the Drosophilidae family. It receives its name because it feeds on fruits under a fermentation process such as apples, bananas, and grapes. It does not constitute an agricultural pest as such, but it is an excellent model organism for the study of insecticidal activity [3,4].

There are several biopesticides with a broad activity spectrum for this purpose, but there is also a global concern about their negative side effects such as ozone depletion (many of them are in halogenated commercial presentations at the shelf), environmental pollution, and toxicity to non-target organisms, pest resistance, and pollutant residues [5]. Due to these latter issues, it is desirable to search for new bioactive compounds against an insect pest attack. Natural products from plant sources often have an advantage over conventional pesticides because they degrade rapidly, and have low toxicity [5].

Chile is surrounded by the Atacama Desert, the Pacific Ocean, the Andes Mountains, and the Antarctic. These natural borders, together with a changing geography and climate, determine great plant and animal biodiversity. For instance, the Chilean flora is composed of approximately 6000 often endemic plant species. In this sense, it should not be surprising that plants have been used since time immemorial to treat diseases of people, animals, and plants (e.g., food and ornamentals). Thus, these backgrounds support our intention to prospectively explore those resources as a new source for biologically-active natural compounds that include environmentally-friendly biopesticides, with low mammalian toxicity and desirable biodegradation features [5,6,7,8]. Therefore, natural products are a good alternative for the development of biopesticides. Many compounds isolated from plant and fungal origin have significant insecticidal activity associated with the presence of alkaloids, iridoids, terpenes, flavonoids, naphthoquinones, anthraquinones, coumarins, phenylpropanoids, and flavonoids [5,6,7,8,9,10,11,12,13].

Since morphine was discovered, more than 12,000 alkaloids have been isolated. About 20% of the species of flowering plants produce alkaloids, and, in each of these species, the alkaloids accumulate in a unique and defined pattern [14,15,16]. However, the investigation on their toxicity to the insect has been largely limited to studies of those compounds found in the Solanaceae family. For instance, the insecticidal properties of nicotine (pyridine alkaloids), are well documented, and its metabolism, and mode of action in the insect, perhaps, are the best understood amidst the great diversity of the naturally-occurring insecticides, leading to the most effective insecticide-derivatives [17].

Benzylisoquinoline alkaloids (BIAs) are a diverse group of biologically active specialized molecules from a large and structurally diverse families of plant secondary metabolites, such as Annonaceae, Araceae, Aristolochiaceae, Berberidaceae, Cactaceae, Combretaceae, Convolvulaceae, Euphorbiaceae, Fumariaceae, Hernandiaceae, Lauraceae, Liliaceae, Magnoliaceae, Menispermaceae, Monimiaceae, Papaveraceae, Ranuculaceae, Rhamnaceae, Rubiaceae, and Rutaceae [17]. The metabolism of BIA is likely of monophyletic origin and involves multiple enzymes yielding structurally diverse compounds [18].

The BIAs are derived from tyrosine that exhibit a wide range of pharmacological activities [14,15,16]. For example, the key intermediate reticuline has been shown to (i) inhibit Ca^2+^ transport by blocking L-type Ca^2+^ channels, resulting in hypotensive and antispasmodic effects [19], (ii) act as a central nervous system depressant being an allosteric modulator of GABA_A_ receptors [20] and (iii) also have weak affinity by nicotinic cholinergic receptors [21].

More complex molecules, such as the bisbenzylisoquinoline alkaloid tetrandrine, have been used to treat autoimmune disorders [22], and hypertension [23], whereas other BIAs demonstrate similar vaso-relaxing properties like reticuline [24]. Among the most widely used BIAs are the analgesics morphine and codeine and the muscle relaxant (+)-tubocurarine [17]. Several BIAs possess defensive properties against pathogenic microorganisms and herbivores. Although not considered essential for normal growth and development, BIAs would most likely play key roles in the defense of plants against herbivores and pathogens [25]. Considering the above, clearly the effects of benzylisoquinolines alkaloids on insect pests have not been well studied. Therefore, one of the main motivations of this study was to elucidate whether BIAs are toxic to adult and larval forms of *D. melanogaster* (Diptera) and *C. pomonella* (Lepidoptera). Therefore, the objective of this study was to understand the acute toxicity, mortality, and insect growth-regulator effects of BIAs. We chose both target insects since the fruit-fly *D. melanogaster* is a good model organism to evaluate insecticidal activity and codling moth *C. pomonella* is an important crop pest. In previous reports, we found that exposure to extracts rich in alkaloids could promote inhibition of growth activity and mediate ecdysone activity [26,27]. In this context, and in order to understand the potential adverse effects of BIAs, we performed a molecular docking study upon ecdysone receptor (EcR), which is very important in regulating the transition from larvae-to-adult in these insects. We also decided to explore the interaction of BIAs with the octopamine receptor (Octβ3R). The octopamine pathway is linked to the activation of ECD receptors since it regulates the synthesis of ecdysone by autocrine signaling. The Octβ3R belong to the family of G-protein-coupled receptor (GPCR) and is orthologous to vertebrate β-adrenergic receptor [28,29]. The activation of this receptor produces an increase in cAMP or release of Ca^2+^. In different types of insect neurons, the second messengers Ca^2+^ and cAMP act as regulators of behavior [30]. In addition, the increase on cAMP or release of Ca^2+^ serves for the activation of several kinases such as PKA and CAMKII that phosphorylate a wide variety of proteins related to the pathway and enzymes involved in the synthesis of ecdysone precursors and 20-hydroxyecdysone [31]. The Octβ3R genetic knockdown produces and arrest in metamorphosis, which also demonstrates the importance of this receptor in the metamorphosis process [32]. In turn, 20-hydroxyecdysone enables an increased activity of tyrosine decarboxylase (TDC), which is the first enzyme responsible for the OA synthesis [33]. All these antecedents also encouraged us to carry out a molecular docking of BIAs on this receptor. Finally, our work aimed to assess the impact of BIA on growth developmental performances and molting development effects in the early life stage of both insect pest species. Even though BIAs clearly form a widespread group of secondary compounds, in the present work BIAs were isolated from different plants that grow in Chile. Some of these plants are *Talguenea quinquenervia* (Rhamnaceae), [26,27], *Discaria chacaye* (Rhamnaceae) [34], *D. crenata* (Rhamnaceae) [35], *D. serratifolia* (Rhamnaceae) [36,37], *Colletia spinossisima* (Rhamnaceae) [38], *Cryptocarya alba* (Lauraceae) [39], and *Peumus boldus* (Monimiaceae) [40]. 

## 2. Results

### 2.1. Phytochemical Analysis

From different plants that grow in Chile, as is detailed in the methodology, four known alkaloids (boldine **1**, coclaurine **2**, laurolitsine **3**, and pukateine **4**) were isolated by conventional methods. After being purified, their structures were determined by classic spectroscopic methods such as nuclear magnetic resonance (NMR), mass spectrometry (MS), and comparison with data reported in the literature (Figure 1 and Figure 2).

*(6aS)-1,10-dimethoxy-6-methyl-5,6,6a,7-tetrahydro-4H-dibenzo[de,g]quinoline-2,9-diol (Boldine*, **1**): m.p.: 160–163 °C. [α]_D_^20^: +121 (c 0.48, MeOH). IR (nujol, cm^−1^): 3532, 3397, 3003, 1466, 1271. ^1^H-NMR (300 MHz, CDCl_3_) δ: 6.64 (s, 1H), 6.83(s, 1H), 7.89 (s, 1H), 3.60 (s, 3H), 3.91 (s, 3H). ^13^C-NMR (75 MHz, CDCl3) δ: 144.4 (C-1), 128.0 (C-1a), 126.2 (C-1b), 150.8 (C-2), 115.6 (C-3), 129.8 (C-3a), 29.1 (C-4), 54.2 (C-5), 63.9 (C-6a), 34.6 (C-7), 130.4 (C-7a), 115.9 (C-8), 147.1 (C-9), 147.8 (C-10), 112.5 (C-11), 124.6 (C-11a). ESI-MS: *m*/*z* (%): 327.96 (calcd. for 328.155), 194.0 (33.1), 205.0 (32.7), 221.9 (26.4), 237.1 (57.1), 264.9 (100), 282.0 (49.8), 297.0 (82.9).

*(1S)-1-[(4-hydroxyphenyl)methyl]-6-methoxy-1,2,3,4-tetrahydroisoquinolin-7-ol (Coclaurine*, **2**): m.p.: 222–224 °C. [α]_D_^20^: + 4.7° (c 0.48, MeOH). IR (nujol, cm^−1^): 3530, 3470, 3300. ^1^H-NMR (300 MHz, CDCl3) δ: 2.70–3.36 (m, 6H), 3.83 (s, 3H), 4.39 (d, 2H, *J* =10.2 Hz), 6.59 (d, 1H, *J* = 4.1 Hz), 6.63 (d, 2H, *J* = 8.4 Hz), 6.79 (s, 1H), 7.02 (d, 2H, *J* = 8.4 Hz). ^13^C-NMR (75 MHz, CDCl3) δ: 108.1 (C-1), 145.8 (C-2), 28.8 (C-4),126.1 (C-4a), 112.7 (C-5), 148.0 (C-6), 145.7 (C-7), 114.0 (C-8), 129.6 (C-8a), 41.7 (C-9), 128.8 (C-10), 131.3 (C-11), 116.6 (C-12), 157.2 (C-13), 116.6 (C-14), 131.3 (C-15). ESI-MS: *m*/*z* (%) = 285.95(0.1) (calcd. for 286.144), 178 (100), 163(20), 107(8).

*(6aS)-1,10-dimethoxy-5,6,6a,7-tetrahydro-4H-dibenzo[de,g]quinoline-2,9-diol (Laurolitsine*, **3**): m.p.: 138–140 °C. [α]_D_^20^: +110 (c 0.69 EtOH). IR (KBr, cm^−1^): 3378, 3285, 2937, 2830, 1582, 1252, 1085, 1010. ^1^H-NMR (300 MHz, CDCl_3_) δ: 6.63 (s, 1H), 6.78 (s, 1H), 7.91 (s, 1H), 3.57 (s, 3H), 3.77 (s, 3H). ^13^C-NMR (75 MHz, CDCl3) δ: 142.7 (C-1), 149.4 (C-2), 114.6 (C-3), 128.8 (C-3a), 125.5 (C-3b), 27.9 (C-4), 42.2 (C-5), 53.2 (C-6a), 35.5 (C-7), 129.1 (C-7a), 115.1 (C-8), 145.9 (C-9), 146.1 (C-10), 112.1 (C-11), 122.9 (C-11a), 125.9 (C-11b). ESI-MS: *m*/*z* (%) = 314.07 (calcd. for 314.139), 165.1 (51.1), 176.1 (15.3), 194.0 (35.9), 205.1 (29.2), 237.1 (54.3), 264.9 (100), 297.1 (90.2).

*(7aR)-6,7,7a,8-Tetrahydro-7-methyl-5H-benzo[g]-1,3-benzodioxolo [6 ,5,4-de]quinolin-12-ol (Pukateine*, **4**): m.p.: 210–212 °C. [α]_D_^15^ −257° (c = 1.0, EtOH). IR (KBr, cm^−1^): 3406, 2969, 1582, 1285, 1043, 895. ^1^H-NMR (300 MHz, CDCl_3_)δ: 2.55 s (3H, NMe), 3.14 dd, *J* = 13.3, 3.4 Hz (lH, H-6a), 5.98 d, *J* = 1.3Hz (lH, OCH20), 6.13 d (*J* = 1.3Hz, lH, OCHzO), 6.65 s (lH, H-3) 6.91 dd (J = 7.2, 1 Hz, lH, H-8 or H-l0), 6.98 dd, (*J* = 6.9, 1 Hz, lH, H-10 or H-8), 7.22 dd (*J* = 8.3, 7.3 Hz, lH, H-9). ^13^C-NMR (75 MHz, CDCl3) δ: 145.7 (C-1), 153.3 (C-2), 114.5 (C-3),129.4 (C-3a), 118.3 (C-3b), 29.4 (C-4), 52.9 (C-5), 62.3 (C-6a), 35.9 (C-7), 128.9 (C-7a), 118.1 (C-8), 139.7 (C-9), 138.2 (C-10), 107.7 (C-11), 120.6 (C-11a), 127.8 (C-11b), 43.9 (N-CH3), 100.2 (O-CH2-O). ESI-MS: *m*/*z* 295 (M^+^, 100%) (calcd. For 295.332), 294 (100), 280 (15), 278 (15), 265 (45), 252 (40), 236 (15), 222 (10).

### 2.2. Insecticidal Activity

#### 2.2.1. Larval Toxicity of the Alkaloids

The results shown in Figure 3 indicate that the four alkaloids tested promoted the death of the larvae of insects used as bioassays. This effect was dose-dependent and time-dependent for both target insects. Analysis of variance (ANOVA) was used to determine the existence of significant differences in the mortality of the test groups with a *p* < 0.05, not observing significant differences, for which it was not necessary to apply a contrast test. Thus, a dose of 50 µg/mL of boldine and pukateine against *D. melanogaster* larvae for the first 24 h of treatment caused an 83% and 70% of death, respectively. At doses of 10 and 30 µg/mL, the alkaloids have a low toxicity during the first 24 h of treatment. This toxicity, expressed in larval mortality, increases significantly over the hours. Thus, after 72 h of treatment, it is possible to observe a mortality above 70% of the larvae with a dose of 10 µg/mL of boldine, laurolitsine, and pukateine. Only coclaurine promotes a lower mortality at this dose. Accordingly, the LD_50_ at 24 h was 78.2, 94.5, 70.8, and 70.9 µg/mL for coclaurine, laurolitsine, boldine, and pukateine, respectively.

Similarly, it can be observed that, at a dose of 50 µg/mL coclaurine, laurolitsine, and pukateine killed more than 80% of the *C. pomonella* larvae after 24 h of exposition. In particular, it should be noted that, at the same dose and at the same time of treatment, boldine only killed 50% of the larvae. An increase in the dose up to 100 µg/mL caused the death of 100% of the larvae within 24 h of treatment.

On the other hand, after 72 h of treatment, an increase in the mortality of the larvae can be observed at lower doses. Thus, coclaurine at 30 µg/mL caused the death of 100% of the larvae. Thus, the LD_50_ at 24 h were 35.4, 30.7, 33.4 and 33.2 µg/mL for coclaurine, laurolitsine, boldine, and pukateine, respectively.

#### 2.2.2. Influence of the Alkaloids on Insect Growth

In this bioassay, we assess the post-ingestion effects of alkaloids on the development of larvae. For this, we used sub-lethal doses of 10 μg/L, since mortality at these doses is null or very low, as previously observed in the acute toxicity study. In a similar way to the acute toxicity test, analysis of variance (ANOVA) was used to determine the existence of significant differences in the mortality of the test groups with a *p* < 0.05, not observing significant differences, for which it was not necessary to apply a contrast test. To begin the study, the first instar larvae of *D. melanogaster* or *C. pomonella* were feed for 24 h with an artificial diet containing the alkaloids. Then, the surviving larvae were changed to a new artificial diet without alkaloids and mortality, growth, pupation and emergence were observed during the life cycle of the insects. The *D. melanogaster* and *C. pomonella* completed development larvae in 10 days and 16 days, respectively. In our study, the effects were monitored for 14 days.

The chronic effects of alkaloids against larval population of *D. melanogaster* are shown in Table 1. During the course of this assay, it was observed that the alkaloids have a chronic toxic effect. This chronic toxicity produced high mortality (Figure 4). Thus, the mortality at the end of assay (14 days) was 100% for laurolitsine **3**, 84% for pukateine **4**, 82% for boldine **1**, and 73% for coclaurine **2**. Additionality, an important effect on surviving larvae development was observed. This effect is manifested in the reduction of both the weight and the length gained. Laurolitsine **3** produces a total blocking of growing of larvae. With boldine **1** the larvae only gained an 11% on weight, whereas, for coclaurine **2** and pukateine, the gains were 43% and 78%, respectively. With regard to the length, the larvae treated with pukateine **4** only gained 8% on length, while the treatments with coclaurine **2** and boldine **1** gained a 73% and 69%, respectively. On the other hand, when compared with the control, the mean time pupation increased between the 2 to 6 days of treatment with boldine **1**, coclaurine **2**, and pukateine **4**. Only a few larvae were able to complete their cycle. Furthermore, the insects that emerged presented significant deformations, mainly of the wings (Figure 5).

In contrast with *D. melanogaster*, a more intense chronic toxic effect was observed for *C. pomonella*, (Table 2) on which all alkaloids induced 100% death upon the larvae at 12 days (Figure 6). After twelve days, all larvae were unviable. During this time, larvae treated with boldine **1** only increase weight in a 7.4% with pukateine **4** in a 34% with laurolitsine **3** in a 46%, and, with coclaurine **2**, in a 99%. With regard to gained length, the effect is similar for each alkaloid. A control larva reaches an average length of 13 to 14 mm and an average weight of 57 ± 1.5 mg.

### 2.3. Docking Study

#### 2.3.1. Octopamine Receptor

A protein model by homology of the Drosophila melanogaster OAMB receptor was obtained through the GPCR-I-TASSER server, according to the experimental procedure detailed by Dacanay et al. [41], where the homologs used for modeling has an identity with OAMB between 18% and 22%.

The docking analysis of octopamine and its receptors is widely used due to the significance of this molecule for the invertebrate biological pathways [41,42]. The structures of the most stable conformations of ligand-protein complexes are presented in Figure 7. The results suggest that boldine (−5.4 ± 0.02 kcal mol^−1^, 109.44 μM) and pukateine (−5.33 ± 0.04 kcal mol^−1^, 123.53 μM) complexes have lower values of predicted binding free energy and inhibition constant, even though pukateine showed an unfavorable interaction between the hydroxyl group of the C-ring and residue TRP 251.

Boldine shows a series of π and π sigma interactions with amino acids Val 111, TRP 251, LEU 254, VAL 191, and ILE 112 in addition to forming a conventional hydrogen bond with residue PHE 179 and two carbon-hydrogen bonds with the residue of GLY 104. In the case of pukateine, in addition to interaction through π and π alkyl bonds with the residues of VAL 191 and ALA 284, the aromatic rings can form stronger bonds of the π sigma type with the VAL 111 residue and a π-sulfur bond with the residue MET 107.

Laurolitsine forms three conventional hydrogen bonds with residues ASN 162, VAL 159, PHE 179, two carbon hydrogen bonds with residues GLY 104, and one π sigma bond with residue VAL 111. Coclaurine forms two conventional hydrogen bonds with ASN 162 and ASN 255 as well as several interactions of the π alkyl type between the aromatic C-ring and the residues LEU 254, VAL 191, and VAL 111. Results from the docking studies suggest that an aromatic motif induces favorable binding.

#### 2.3.2. Ecdysteroid Receptor

The deformations in the wings of adult insects of *D. melanogaster* as well as an elongation in the mean pupation time suggest a potential interaction with the ecdysteroid receptor (EcR) in insects. Certain molecules that binds to this nuclear receptor can induce lethal molts in all larval stages of insect [43]. Therefore, we thought it is important to study a possible interaction of the alkaloids under study with EcR. The results of molecular docking shown that at least 8 residues in the ligand binding domain (LBD) were involved in the interactions between the alkaloids and the EcR (Figure 8). Boldine could form polar hydrogen bonds with Met 380 in the LBD interacting with the hydroxyl group. Meanwhile, the benzene ring of boldine formed the π-π interaction with the residues Met380, Met381, and Ile339. The hydrophobic zone of boldine was connected in the LBD by alkyl and Pi-alkyl bonds with the hydrophobic residues Ile339, Tyr408, Val416, Val384, and Leu420. Furthermore, it is possible to observe a non-conventional interaction carbon-hydrogen between the carbon backbones of the ligand with Ile339. By the other hand, coclaurine could form a polar hydrogen bond with Met380, Arg383, and Glu309. The benzene ring formed π-π interaction with Met342, Ala398, Pro 311, and Val384. A non-conventional carbon-hydrogen interaction was formed with Ile339, and a Pi-donor hydrogen bond between Phe397 and hydrogen of the amino group.

The 3D interaction and binding modes of boldine and coclaurine within the LBD provided detailed structural insights into the interaction between the assayed compounds and the receptor. The formation of hydrogen bonds in the ligand-receptor complex, the hydrophobic and π-π interaction between the compounds, and the ecdysone receptor EcR, which played key roles in promoting the binding affinity of the compound to regulate and disrupt the larvae growth. That could lead to and promote the death of the insects observed.

## 3. Discussion

Plants use different defense mechanisms against the attack of other organisms. Most of them use certain types of compounds among a variety of secondary metabolites that can be produced, such as acetogenins, alkaloids, phenylpropanoids, steroids, and terpenoids. In this way, they repel predators and control competitive organisms in their environment. Several alkaloids have marked pathological effects on insects and their endosymbionts. Thus, depending on the dosage received, these activities could manifest as retarding of growth, development, and reproduction as well as paralysis and mortality.

There is good evidence that the toxic effects of nicotine and related alkaloids are due to an interaction of those alkaloids with the acetylcholine receptor, which is an integral membrane protein belonging to the ion channel type receptor family that binds acetylcholine and mediates the transmission of the neuron impulse.

Within Chilean flora, many species of plants growing mainly in the south-central region of the country are known because the extracts obtained from its different botanical parts not only have medicinal value but also could be used as insecticides and parasiticides [27]. Among these native resources, from barks or aerial parts from plants of Lauraceae, Rhamnnacae, and Monimiaceae families, several BIAs have been isolated. For instance, boldine 1, coclaurine 2, laurolitsine **3**, and pukateine **4** were successfully isolated in high purity from the aerial part or bark from *T. quinquenervia*, *D. chacaye*, *C. alba*, and *P. boldus*. Toxic and insect growth regulator (IGR) properties of compounds **1**–**4** were evaluated against larvae of fruit fly *D. melanogaster* and lepidopteran crop pest *C. pomonella*. These compounds show acute dose-dependent toxicity with DL_50_ 33.16 µg/mL for *C. pomonella* and 78.6 µg/mL for *D. melanogaster*, respectively. On the other hand, the experimental results regarding effect on insect growth showed that alkaloids disrupted the insect development, exhibiting a decrease in the weight and length gained, and the pupation time elongation. Additionally, compared with typical imagoes we observed that few insects that hatched (*D. melanogaster*) presented wing deformations (Figure 5). It is known that alterations in the growth development of this insect can be ascribed to nutrition problems or alteration in its live cycle (metamorphosis) [44]. The first alternative can be discarded because the control does not display morphological alteration. Therefore, the alterations observed can only occur due to the interaction of the alkaloids with the insect hormonal system. In fact, our in-silico study showed that these alkaloids could be interacting with the heterodimer Ecdysone receptor. In line with this statement, our first approach via molecular docking shows that boldine **1** and coclaurine **2** have the highest score for binding energy to the ecdysone receptor. Natural Product (NP)-based agonists for ecdysone receptor, also known as moulting-accelerating compounds (MACs), are considered to be a selective group of insecticides, and constitute a new approach in pest management. Thus, molecules that present this capacity are of great interest. As mentioned, studies on the insecticidal activity of alkaloids of type BIAs are scarce. It is described that anonaine **5**, which is present in members of the plant families Magnoliaceae and Annonaceae, acts as an effective insecticide against aphids with activity comparable to rotenone [45,46]. Similar alkaloids have shown larvicidal and development regulatory activities for the malaria vector *Anopheles stephensi* [47]. Albeit, information about the insecticidal mode of action of these compounds is lacking. Some alkaloids like anonaine have a smooth muscle relaxant effect by modulation of adreno-receptors and voltage-gated Ca^2+^ channels in rats [19]. Other works report that isoboldine **6** inhibit the feeding of *Spodoptera* sp. larvae at 200 µg/mL [48]. Studies in vertebrate treated with BIAs show that they are capable to modulate adreno-receptors, voltage-gated Ca^2+^ channels, and inhibit nicotinic acetylcholine receptor (nACh) signaling [21,49,50]. Heng et al. [51] has reported that isolaureline **7** and dicentrine **8** act as antagonists of 5-HT2 and α1 receptors. R and (*S*)-glaucine **9** are also antagonized as the α1 receptor, but they behaved very differently from other compounds at the 5-HT2 receptor. While (*S*)-glaucine **9** acted as a partial agonist at all three subtypes of 5-HT2 receptors, (*R*)-glaucine **10** seems rather acting as a positive allosteric modulator at the 5-HT_2A_ receptor. In the case of insects, the metamorphosis process is regulated by the ecdysone receptor in representatives of the genera *Drosophila*, *Bombix*, *Chiro*, *Bactrocera*, *Nicrophorus*, among many others [52,53]. In turn, this receptor receives autocrine chemical signals such as that of the steroidal hormone 20-hydroxy-ecdysone. This hormone is synthesized from the cholesterol present in the diet of insects inside the prothoracic gland, which is a process regulated by the expression of several ecdysteroidogenic enzyme genes during the larva-to-pupal transition [32].

In the parathoracic gland, the synthesis of ecdysone is controlled by the activation of octopamine receptors. Octopamine (OA), tyramine (TA), dopamine (DA), serotonin (5-HT), and histamine (HA) are known to be biogenic amines that affect diverse physiological and behavioral processes in invertebrates. As mentioned in the introductory section, recent findings indicate that an additional adrenergic system exist in at least some invertebrates, where OA and its precursor TA have replaced the neurotransmitters noradrenaline and adrenaline [9,54]. Hence, in a second in silico approach, we performed molecular docking of BIAs on the protein model of Octβ3R obtained through homology of the *D. melanogaster* OAMB receptor. Among the set of BIAs assayed in this work, boldine and pukatein displayed the most promissory binding-free energy values. To date, the potential of BIAs for the development of insecticides that act on the metamorphosis of the insect by interfering simultaneously with the synthesis of ecdysone and octopamine signaling has not been explored. Although alkaloids of this type have been studied in different contexts, a review of the available literature suggests that very little is known about their effect on insects. In a study by Bianchini et al. [54], adult specimens of *D. melanogaster* were exposed to a food with 3 mM of manganese and aqueous extract of *P. boldus* (5 mg/mL) or boldine (327.37 µg/mL). Boldo extract reduced mortality and Mn-induced oxidative stress. Additionally, the extract reversed the motor dysfunction induced by Mn. However, these effects could not be attributed to the presence of alkaloids such as boldine since the aqueous extract of *P. boldus* were always more active than the single pure substance. In another study, the inhibitions of copper induced-toxicity in *D. melanogaster* was demonstrated for extracts of *P. boldus* [55]. This protection was shown by an increase in mRNAs associated with antioxidant enzymes (catalase, superoxide dismutase, thioredoxin reductase), transcription factors such as nuclear factor erythroid 2-related factor 2 (Nrf2), and increased mRNA levels of acetylcholinesterase. Again, these effects underlie to the presence of polyphenols rather than alkaloids in the aqueous extract. Regarding pukatein, it is known that, like boldine, it has antioxidant and dopaminergic properties [56].

The physiological roles of these amines in invertebrates is associated with a variety of processes including epithelial transport, olfaction, reproduction, flight, and locomotion [57,58]. Furthermore, it is known that, when OA binds to its β3-octopamine receptor (Octβ3R), the synthesis of ecdysone [59,60] is stimulated. This receptor belongs to the family of G-protein-coupled receptors (GPCR) and is orthologous to vertebrate β-adrenergic receptors [28,29]. The Octβ3R genetic knockdown produces an arrest in metamorphosis, which demonstrates the importance of this receptor in the metamorphosis process [45]. Moreover, 20-hydroxyecdysone enables an increased activity of tyrosine decarboxylase (TDC), which is the first enzyme responsible for the OA synthesis [33]. In different types of neurons, the activation of Octβ3R leads to an increase in the levels of the second messengers Ca^2+^ and cAMP, which act as regulators of behavior in insects [30]. Different studies on these receptors indicate that they can be blocked by some known drugs. So far, phentolamine, mianserin, cyproheptadine, and promethazine have been reported as the most potent inhibitors of this receptor [61,62]. In *D. melanogaster*, the dopamine receptors (DOP1, DOP2, and DOP3) are paralog with Octβ3R and both can be inhibited by mianserine, epinastine, and spiperone, even though the inhibition of DOP receptors occurs at lower concentrations than those found for Octβ3R [41,63]. Recently, an interesting crosstalk between dopamine receptors and ecdysone pathway was reported by Kang et al. In this study, 20-hydroxy-ecdysone binds to dopamine receptors stimulating arrests larval feeding and promoting pupation [64]. This inhibition of dopamine function decrease motor function in *D. melanogaster* and also interfere with the reward-motivated behaviors by causing a decrease in feeding.

The information provided by the literature on the pharmacological properties of BIAs as well as the results obtained from the acute and chronic effect of these alkaloids against both types of the larvae tested, lead us to propose that these alkaloids could also interact with the dopaminergic system present in insects. As a result of the failure in the displacement of the larvae in the feeding medium, particularly in *D. melanogaster* (construction of tunnels), an alteration in the feeding behavior of the larvae is produced. A similar effect was previously reported by Wada and Munakata [65] in larvae of *Spodoptera*. This effect would be related to the ability of BIAs to block Octβ3R and voltage-gated Ca^2+^ channels [66,67]. Therefore, we can infer that the BIAs would replicate their action in insects through a direct interaction with the octopaminergic system, affecting their behavior. The effect of BIAs upon metamorphosis could be linked to an indirect mechanism involving the inhibition of the ecdysteroids synthesis.

## 4. Materials and Methods

### 4.1. Equipment and General Experimental Procedures

All the chemicals were obtained from Merck (Darmstadt, Germany). ^1^H-NMR and ^13^C-NMR, COSY, NOESY, HMQC, and HMBC spectra were recorded on a Bruker spectrometer 400 MHz using CDCl_3_ as the solvent and TMS as the internal standard. Chemical shifts were reported in units of (ppm) and coupling constants (J) expressed in Hz. Column chromatography (CC) was performed on Silica Gel 60 (40–63 µm, Merck, Darmstadt, Germany). TLC was carried out on pre-coated Silica Gel 60 GF254 plates (Merck). Spots on chromatogram were visualized under a UV light and by spraying with 5% H_2_SO_4_ in methanol, and then heating at 110 °C for 5 min. Preparative TLC was conducted with glass precoated Silica Gel GF254 (Merck, Darmstadt, Germany).

### 4.2. Plant Material

*Talguenea quinquenervia* (Gill. et Hook) Johnst. aerial parts was collected on the roadside at a pass 4.7 km NW of Portezuelo on the road to Ninhue (36°34.105′ S, 72°26.865′ W), XVI Region, Chile, in June 2017. Voucher specimens have been deposited in the Herbarium of the Basic Science Department, University of Bio-Bio (Voucher DS-2010/05-16246) and the Herbarium of the University of Illinois, at Urbana-Champaign, Illinois, USA, (ILL, Voucher DS-16246). *Discaria chacaye* (Vent.) B.&H. ex Master was collected on the road to Yungay (37°06′57″ S, 72°15′25″ N), XVI Region, Chile, in June 2017. Voucher specimens have been deposited in the Herbarium of the Basic Science Department, University of Bio–Bio, Chile and Herbarium of the University of Illinois, at Urbana-Champaign, Illinois, USA, (ILL, Voucher DS-16253). *Peumus boldus* (Molina) was collected at a Diguillin river, XVI Region, Chile, in the summer of 2017. Voucher specimens have been deposited in the Herbarium of the Basic Science Department, University of Bio–Bio Bio. *Cryptocarya alba* (Mol.) Looser was collected near the Chillan city in Summer of 2017. Voucher specimens have been deposited in the Herbarium of the Basic Science Department, University of Bio-Bio, Chile.

### 4.3. Extraction and Isolation

Aerial parts of plants in the study (500 g) were exhaustively extracted with MeOH in a Soxhlet apparatus for 12 h. The resulting MeOH extract was filtered and concentrated under vacuum to obtain a crude residue (139 g). The total extract was dissolved in water (50 mL) and acidified to pH 2. The acidic solution was exhaustively extracted with diethyl ether (Et_2_O) (5 × 50 mL) to yield an acidic ether extract. The aqueous solution was made alkaline to pH 10 with ammonium hydroxide (NH_4_OH) and re-extracted with Et_2_O. This was followed by washing with distilled water, dried over anhydrous sodium sulphate, and evaporated to give an alkaloid fraction. Purification of alkaloids was performed by column chromatography and thin layer preparative chromatography. Fractions of 10–15 mL were collected, monitored by TLC (UV 254 nm, Dragendorff’s reagent, Merck, Darmstadt, Germany), and combined according to their profiles. Identification of alkaloids was performed by the spectroscopy method (IR, NMR, and MS) and comparison with a standard sample.

### 4.4. Insect Bioassays

Fruit-Fly (*D. melanogaster*, Diptera) pupae were collected from infected fruit and placed in bottle glass with an artificial diet, which contains brewers’ yeast (60 g), glucose (80 g), agar (12 g), propionic acid (8 mL), and water (1000 mL) under a controlled condition (16:8 h light-dark photo-period at 22/24 °C and 80% relative humidity). Codling moth (*C. pomonella* L., Lepidoptera: Tortricidae) pupae were collected from infected apple tree in agricultural production at an experimental field. The pupae were placed in a plastic box under controlled conditions (16 h:8 h light-dark photoperiod at 26/24 °C and 60% relative humidity). Upon emergence, the adults were transferred to a paper cylinder with moist cotton and kept under the same conditions as the pupae. Neonate larvae were allowed to feed on an artificial diet, which contains soya flour, sucrose, skimmed milk powder, dry yeast, sunflower oil, cholesterol, choline chloride, ascorbic acid, Vanderzant vitamin, Wesson’s salt, methyl hydroxybenzoate, sorbic acid, water, formalin, ethanol, and agar. Unfed first-instar larvae were used for bioassays.

### 4.5. Bioassay for Insecticidal Activity against Larvae of D. melanogaster

The bioassay was carried out as described by Miyazawa et al. [68] with an adjustment of the amount of food given (10 mL instead of 5 mL). Artificial diet was mixed with ethanol-compound (maximum value: 50 µL of ethanol) at concentrations of 10, 30, 50, and 100 µg mL^−1^. Two negative controls were performed: one control containing only the artificial diet and another one containing 50 µL of ethanol. An artificial diet (10 mL) containing the respective compound were placed in a Petri disk. Ten second-instar larvae of *Drosophila melanogaster* was placed in each Petri disk. Five replicates for each treatment were carried out. The Petri disks were placed under controlled conditions in a cultivation chamber. The insecticidal activity of extracts was determined by measuring mortality periodically for which the number of live larvae was counted every 24 h for three days. The LD_50_, which is the concentration that produces 50% mortality, was determined by log-probit analysis.

### 4.6. Bioassays for Insecticidal Activity against Larvae of C. pomonella

The bioassay was carried out as described by Piskroski and Dorn [69] with an adjustment of the amount of food given (10 mL instead of 5 mL). An artificial diet was mixed with the ethanol-compound (maximum value: 50 µL of ethanol) at concentrations of 10, 30, 50, and 100 µg mL^−1^. Two negative controls were performed: one control containing only the artificial diet and another one containing 50 µL of ethanol. An artificial diet (10 mL) containing respective compound were placed in a Petri disk. Ten third-instar larvae of *Cydia pomonella* was placed in each Petri disk. Five replicates for each treatment were carried out. The Petri disks were placed under controlled conditions in a cultivation chamber. The insecticidal activity of extracts was determined by measuring mortality periodically for which the number of live larvae was counted every 24 h for three days. The LD_50_, which is the concentration that produces 50% mortality, was determined by log-probit analysis.

### 4.7. Growth Regulation Assay

Ten newly hatched larvae of *D. melanogaster* or *C. pomonella* are placed in a Petri dish containing artificial diet and the substance to be tested at the lowest concentration (10 µg/mL). Mortality is determined, and surviving larvae are transferred to a Petri dish with fresh food without the substance to be tested (alkaloids). Every 24 h, live larvae, growth, weight, and pupation are counted. Five replicates were performed. The growth index (GI) was calculated according to Zhang et al. [70].

### 4.8. Ligand Construction and Docking Studies on the Ecdysone Receptor EcR and OAMB Receptor

#### 4.8.1. Receptor Modeling and Ligand Preparation

The amino acid sequence to make the OAMB receptor was obtained from the UniProt database (ID: Q7JQF1-1). Since the crystal structure of the receptor has not been isolated, a homology model obtained from the GPCR-I-TASSER platform was used [71]. With this platform, a rational model of the structure in 3D was obtained. GPCR-I-TASSER automatically selects the putative templates by threading through their GPCR PDB library, which is followed by template-based fragment assembly to construct a full-length model [40,72]. For the ecdysone receptor, the crystal structure of *H. virescens* (PDB 2R40, 2.4 Å resolution) obtained from Protein Data Bank (http://www.rcsb.org) was used.

Ligand 3D structures were prepared using Spartan’10 1.1.0 2011 software and their geometry were optimized using Density Functional B3LYP **G ab initio methods *in Vacuo*.

#### 4.8.2. Molecular Docking

Molecular docking simulation was carried out using the AutoDock 4.0 software package [43]. The Discovery Studio 4.0 (BIOVIA Discovery Studio, 2016) was used for the preparation of the protein for docking simulation. Water and other small molecules were removed. In order to add polar hydrogen atoms, the Hydrogen module in AutoDockTools (ADT) graphical interface was used. The Kollman united atom partial charges were assigned for the proteins. The Lamarckian Genetic Algorithm (LGA) was employed for protein-ligand rigid-flexible docking. Octopamine was selected as the positive control in the molecular docking studies. Using the binding affinity and the binding site of OA as a reference, the potential of other ligands is shown. Grid maps were computed using AutoGrid in the grid box of dimensions 60 × 60 × 60 with 0.375 Å spacing.

### 4.9. Statistical Analysis

Data are average results obtained by means of five replicates and independent experiments and are presented as average ± standard errors of the mean (SEM). Data were subjected to analysis of variance (ANOVA) with significant differences between means identified by GLM procedures. Results are given in the text as probability values, with *p* < 0.05 adopted as the criterion of significance. The LD_50_ calculated by PROBIT analysis based on the percentage of inhibition obtained at each concentration of the samples. LD_50_ is the concentration that produces 50% mortality. Complete statistical analysis was performed by means of the Micro-Cal Origin 6.1 statistical and graphs PC program.

## 5. Conclusions

We obtained four purified benzylisoquinoline alkaloids (coclaurine, laurolitsine, boldine, and pukateine). These compounds were isolated from selected Chilean Rhamnaceae plants species and evaluated their insecticidal activity against the insect pest species. The results showed that these alkaloids possess acute and chronic insecticidal effects against *D. melanogaster* and *C. pomonella*, including alteration of the feeding, deformations, failure in the displacement of the larvae in the feeding medium of *D. melanogaster*, and mortality of the visible effects. The docking evaluation of the interaction with the ecdysone receptor show that boldine and coclaurine have the highest binding energy to the ecdysone receptor. Therefore, it can infer that the alkaloids used in this study would replicate their action through the interaction with the octopaminergic system of insects.

These compounds are major constituents of several reputed herbal drugs, particularly those of the family Rhamnaceae. Anti-cancer and anti-inflammatory activities from several plants of this family have been reported. However, the biocide potential of these family plants looking for control of insects on crops has few studies. The development of new, more innocuous, and environmentally-friendly insecticides remains a challenge.

One of the targets of action of the insecticides mainly focus on the inhibition of acetylcholinesterase, and ionic channels among others. Therefore, during recent years, the dopaminergic system of insects is seen as a target for the development of a new type of insecticide. In this sense, knowledge about the effects of BIAs on insects can be useful for the design of new, more specific, harmless, and environmentally-friendly insecticides.

## Figures and Tables

**Figure 1 molecules-25-05094-f001:**
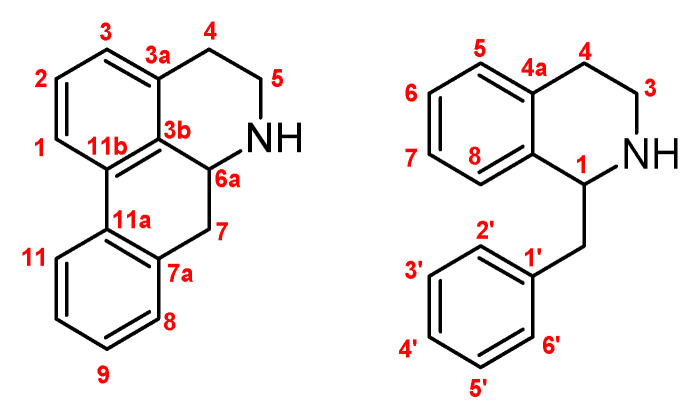
Numbering used for aporphines (**left**) and tetra-hydro-isoquinolines (**right**).

**Figure 2 molecules-25-05094-f002:**
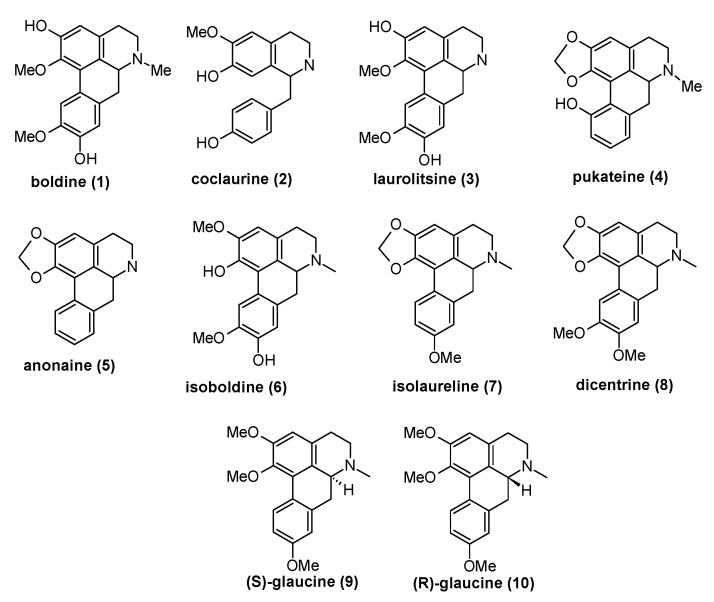
The structures benzylisoquinoline types **1**–**4** of alkaloids isolated from *T. quinquenervia*, *D. chacaye*, *C. alba*, and *P. boldus*, and the other alkaloids mentioned in the text.

**Figure 3 molecules-25-05094-f003:**
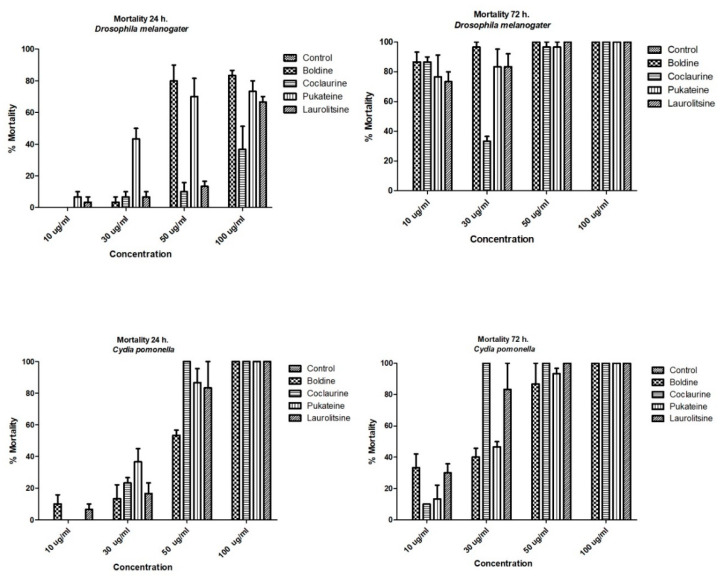
Mortality of benzylisoquinoline alkaloids (BIAs) against *D. melanogaster* L2 and *C. pomonella* L3.

**Figure 4 molecules-25-05094-f004:**
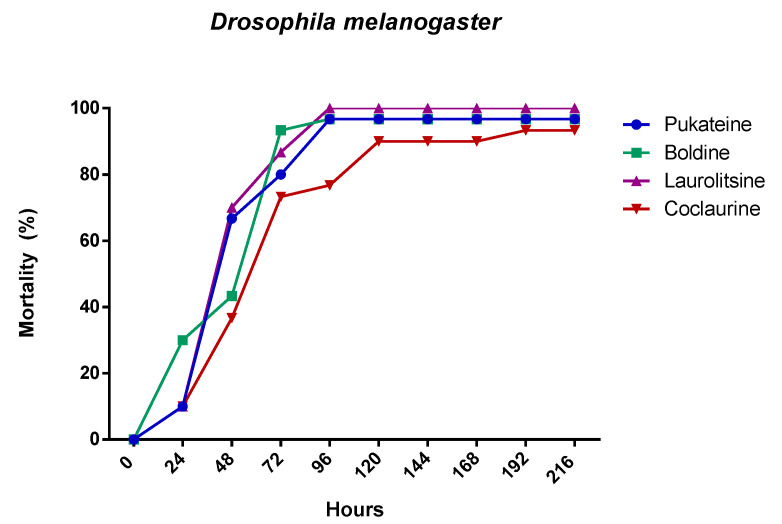
Mortality evolution of *D. melanogaster* after 24 h of treatment with 10 µg/L of the alkaloids.

**Figure 5 molecules-25-05094-f005:**
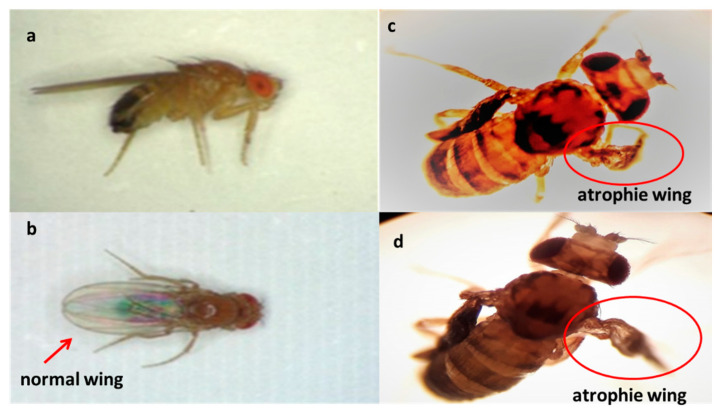
Deformations observed in adult insects of *D. melanogaster*. Normal insect (**a**,**b**) pukateine effect (**c**) and boldine effect (**d**).

**Figure 6 molecules-25-05094-f006:**
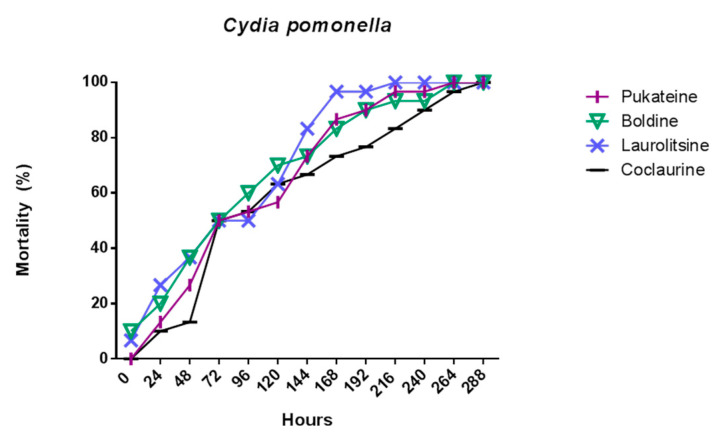
Mortality evolution of *C. pomonella* after 24 h of treatment with 10 µg/L of the alkaloids.

**Figure 7 molecules-25-05094-f007:**
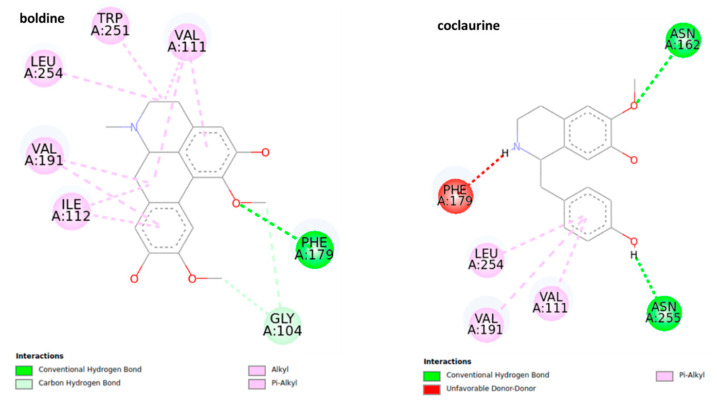
2D representation for the ligand interaction diagram of the OAMB-ligand complex obtained by molecular docking of BIAs.

**Figure 8 molecules-25-05094-f008:**
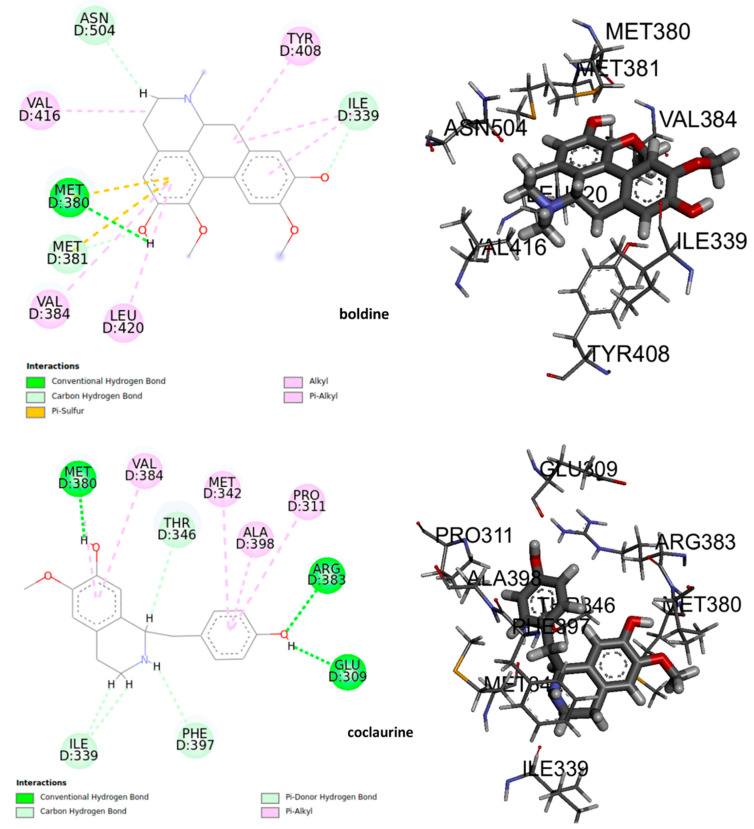
The active site of the complex ligand – ecdysone receptor, 2D and 3D structures.

**Table 1 molecules-25-05094-t001:** Activity of alkaloids **1**, **2**, **3**, and **4** on complete life cycle: growth, pupation, and emergence parameters of *D. melanogaster* (after 14 days of incubations).

Treatment	Mortality after 24 h (%) 10 µg/mL	Cumulative Mortality ^a,b,c^ (%)	Weight Gained (%)	Length Gained (%)	Pupation %	Mean Pupation Time (days)	% IG	Emergence (%)
Control	0	0	100	100	100	8	100	100
Boldine **1**	0	82.4 ± 3.4 ^a,b,c^	11.2 ± 0.07	68.6	3.3	14	3.3	3.3
Coclaurine **2**	0	73.0 ± 3.4 ^a,b,c^	42.9 ± 0.06	72.9	6.6	12	6.6	6.6
Laurolitsine **3**	3.3	100 ± 0.0 ^a,b,c^	0.0 ± 0.0	0	0	0	0	0
Pukateine **4**	10	83.8 ± 2.1 ^a,b,c^	77.5 ± 0.03	7.9	3.3	10	3.3	3.3

^a^ Mean of five replicates, ^b^ Means followed by the same letter within a column ± SE values are not significantly different in an ANOVA test at *p* ˂ 0.05 (treatments are compared by concentration to control), 95% confidence limits. ^c^ cumulative mortality.

**Table 2 molecules-25-05094-t002:** Activity of alkaloids **1**, **2**, **3**, and **4** on complete life cycle: growth, pupation, and emergence parameters of *C. pomonella* (after 14 days of incubations).

Treatment	Mortality after 24 h (%) 10 µg/mL	Cumulative Mortality ^a,b,c^ (%)	Weight Gained (%)	Length Gained (%)	Pupation %	Mean Pupation Time (days)	% IG	Emergence (%)
Control		0	100	100	100	8	100	100
Boldine **1**	10	100 ± 0.0 ^a,b,c^	7.40 ± 0.07	68.3 ± 0.07	0.0	-	-	-
Coclaurine **2**	0	100 ± 0.0 ^a,b,c^	98.5 ± 0.06	71.4 ± 0.06	0.0	-	-	-
Laurolitsine **3**	6.7	100 ± 0.0 ^a,b,c^	46.0 ± 0.07	66.7 ± 0.07	0.0	-	-	-
Pukateine **4**	0	100 ± 0.0 ^a,b,c^	33.6 ± 0.20	66.3 ± 0.08	0.0	-	-	-

^a^ Mean of five replicates, ^b^ Means followed by the same letter within a column ± SE values are not significantly different in a ANOVA test at *p* ˂ 0.005 (treatments are compared by the concentration to control), 95% confidence limits. ^c^ cumulative mortality.

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
