# Peer review of "Assessment of Insecticidal Activity of Benzylisoquinoline Alkaloids from Chilean Rhamnaceae Plants against Fruit-Fly Drosophila melanogaster and the Lepidopteran Crop Pest Cydia pomonella"

_molecules, 2020, doi:10.3390/molecules25215094_

Round 1

Reviewer 1 Report

This is a revised version of the manuscript entitled "Assessment of Insecticidal Activity of Benzylisoquinoline Alkaloids from Chilean Rhamnaceae Plants against Fruit-Fly Drosophila melanogaster and the Lepidopteran Crop Pest Cydia pomonella".

The authors have done efforts to improve their manuscript. There is still few points that can be perfected:

Why the word “apophines” appears in keywords and is not present in the abstract?

P5 line 169 Figure 2, be more precise in the legend and indicate larval stage of the insect if no it can be adults.

P6, Table 1, please be precise, “mortality (%) 10 µg/L” after how many hours, days or whatever? Give the complete definition of IG %. Same remark for Table 2.

P7 line 225, the authors still did not make links between the different parts of the manuscript. Please give explanation about the fact to look at the octopamine receptor. The authors have done it for ecdysone receptor, why not for octopamine receptor?

Figure 6 is too small, not readable. Same remark for figure 7.

P10 line 320, correct Bombix by Bombyx.

Reviewer 2 Report

General comments

This paper reports results on the activity of benzylisoquinoline from Chilean Rhamnaceae plants against Drosophila melanogaster and the Cydia pomonella. The Authors made some changes to the manuscript however, in my opinion, the corrections of the text are insufficient.

The results are interesting however, no statistical tests were presented in the paper therefore the data provided are not solid, statistically stable, and controlled. That is why it is not possible to write appropriate conclusions.

It is true that the Authors wrote that the differences among treatments were identified by ANOVA or/and Student–Newman–Keuls test, however, throughout the text, there is no reference to the obtained statistical differences. In the caption of Tables 1 and 2, there is information about the statistical test used, however, I cannot see either in the table or in the text an appropriate commentary to the obtained differences. In addition, the Authors describe in Materials and Methods that they used two tests. I don't know what the reason was and what test was finally used.

The quality of the figures 6 and 7 should also be improved, as they are completely unreadable

That's why I think the article should not be published in a present form.

Specific comments:

L39: please remove double spaces

L43: please remove a dot before the bracket

L104: please remove a dot before Cryptocarya

Table 1 and Table 2: “a Mean of five replicates; b Means followed by the same letter within a column ± SE...” I cannot see any “a” or “b” letter...

Fig. 5 – I don't see any deformations. How should the D. melanogaster look without deformation? Please mark the difference with an arrow or delete this figure.

Fig. 6 and 7. Please correct Fig. 6 and 7 because the font is illegible.

The sequence of the figures described is not chronological, L182 - Fig. 3, L192 - Fig. 5, L206 - Fig. 4.

There is no reference to statistical analysis in the description of whole results!

I can read that: “Data were subjected to analysis of variance (ANOVA) with significant differences between means identified by GLM procedures. Differences between treatment means were established with a Student–Newman–Keuls 465 (SNK) test.” Then what test was finally done? Both? For all analyzes? Please explain it to me.

Round 2

Reviewer 2 Report

Review molecules-966501 - Assessment of Insecticidal Activity of Benzylisoquinoline Alkaloids from Chilean Rhamnaceae Plants against Fruit-Fly Drosophila melanogaster and the Lepidopteran Crop Pest Cydia pomonella by Quiroz-Carreño et al.

My main concerns have been addressed. I believe that the manuscript fits into the Journal’s aims and scope, and I think it is interesting enough to be published in the present form. Congratulations!